



# Decoupled simulations of offshore wind turbines with reduced rotor loads and aerodynamic damping

Sebastian Schafhirt, Michael Muskulus

Department of Civil and Environmental Engineering, Norwegian University of Science and Technology, Trondheim, 7491,
Norway

Correspondence to: Sebastian Schafhirt (sebastian.schafhirt@ntnu.no)

**Abstract.** Decoupled load simulations are a computationally efficient method to perform a dynamic analysis of an offshore
wind turbine. Modelling the dynamic interactions between rotor and support structure, especially the damping caused by the
rotating rotor, is of importance, since it influences the structural response significantly and has a major impact on estimating
fatigue lifetime. Linear damping is usually used for this purpose, but experimentally and analytically derived formulas to
calculate an aerodynamic damping ratio often show discrepancies to measurement and simulation data. In this study
decoupled simulation methods with reduced and full rotor loads are compared to an integrated simulation. The accuracy of
decoupled methods is evaluated and an optimization is performed to obtain aerodynamic damping ratios for different wind
speeds that provide the best results with respect to variance and equivalent fatigue loads at distinct output locations. Results
show that aerodynamic damping is not linear, but that it is possible to match desired output using decoupled models.
Moreover, damping ratios obtained from the empirical study suggest that aerodynamic damping increases for higher wind
speeds.

## 1 Introduction

The simulation of an offshore wind turbine (OWT) in time-domain under combined aerodynamic and hydrodynamic loading
is currently considered as the most accurate method to analyse the support structure dynamics and forms the basis for
interpretation of the characteristic combine load effect (DNVGL-ST, 2016). However, it is still computationally demanding
and requires special simulation software. Conceptual or parameter studies during the design process (Arany et al., 2017) or
for research purpose (Cheng et al., 2012) usually necessitate to perform numerous simulations, thereby leading to a time
consuming task. These simulations are, therefore, often carried out with different analysis methods, such as frequency-
domain calculations (van der Tempel and de Vries, 2005; Ziegler et al., 2015) or substructuring techniques (van der Valk and
Rixen, 2012) and/or simplified/reduced models (Muskulus and Schafhirt, 2015). For most of the reduced models the
aerodynamic loading is simplified by removing the rotor-nacelle-assembly from the support structure and replacing the aero-
elastic computation with precomputed or stochastic generated rotor loads acting as a point force or moment at tower top
(Dong et al., 2011; Abhinav and Saha, 2015; Kim and Lee, 2015; van der Male and Lourens, 2015; Schløer, et al., 2016; Ong



et al., 2017) The main advantages are the faster simulation time, since the aero-elastic computation rather is a time-consuming task, and the possibility to use standard finite-element or multi-body software (Muskulus and Schafhirt, 2014). It has been shown that the use of rotor load time series combined with an efficient substructuring technique (van der Valk and Rixen, 2012) can speed up the dynamic analysis for a commercial support structure design by a factor of 375. Harnessing the

power of Graphics Processing Units and performing the computation in parallel results in an additional twelve times faster execution time thereby accelerating the dynamic analysis by a factor of 4722 compared to a time-domain simulation including aero-elastic calculations (Schafhirt et al., 2015).

Additional reasons to simplify the aerodynamic loading on a support structure are for example when solely the response of the support structure is of interest or turbine data are not available. The latter is usually the case when support structure

designer and wind turbine supplier do not share detailed data during the design process.

The main drawback of using a model with simplified aerodynamic loading is the missing modelling of interactions between the rotor and the support structure. The most important interaction is the so-called aerodynamic damping, which refers to the effect that the vibration of the support structure is damped by the rotor (van der Tempel, 2000; Kühn, 2001). Especially in operational cases, when the rotor is rotating, aerodynamic damping significantly contributes to the total damping. Damping

has a major impact on the fatigue of an OWT and since fatigue is usually a design driving criterion in dimensioning the support structure, a proper representation of aerodynamic damping is crucial for the dynamic analysis of support structures for OWTs (Muskulus and Schafhirt, 2014).

The most common practice to include aerodynamic damping in simplified models or analyses methods (e.g. frequency-domain calculations) is the application of a discrete dashpot at tower top for time-domain simulations (Schløer, et al., 2016;

Ong et al., 2017) or to add an additional aerodynamic damping value to the damping in the transfer function for frequency-domain calculations (Salzmann and van der Tempel, 2005). Damping values for the transfer function or a damping coefficient for a discrete dashpot are derived in different ways. Often these values are empirically obtained from simulation or measurement data, theoretically from analytical derived formulas or a constant value is assumed for all wind speeds.

Garrad (1990) derived a formulation for aerodynamic damping of constant speed wind turbines, which was later extended by

Kühn (2001), who also proposed numerical linearization and non-linear time-domain simulations to derive aerodynamic damping values. The latter method calculates the aerodynamic damping ratio from a free vibration response of the tower top after an impulse is applied. Garrad's derivation, Kühn's closed-form model and an additional method proposed by van der Tempel (2000) were compared and showed good agreement for operational wind speeds (Salzmann and van der Tempel, 2005). However, these methods are applicable only on constant speed turbines.

Salzmann and van der Tempel extended the methods and derived a formulation for variable speed wind turbines. Van der Tempel's method requires time-domain simulations since the damping ratio is computed by considering changes in thrust due to variations in wind speed. Alternatively, they presented a second method based on an analytical formula. It has been shown that these two approaches for variable speed wind turbines do only match for below rated wind speed and show different behaviour above rated wind speed. Furthermore, both methods do not match damping ratios that were empirically





determined from frequency-domain calculations. This was done by increasing the aerodynamic damping until the mudline
bending stress response spectra of the frequency-domain calculations matched the response spectra of a 7-hour time-domain
simulation (Salzmann and van der Tempel, 2005).

Hansen et al. (2006) used a similar approach as proposed by Kühn (2001). They estimated modal damping for the first fore-
aft and side-side mode of a 2.7 MW turbine under operating conditions from the transient decay of the turbine after exciting
it with its natural frequency. The results are compared with another experimental method, an operational modal analysis
based on stochastic subspace identification (SSI). The SSI method estimates the modal damping and only requires time-
series of the dynamic response of the operating wind turbine due to ambient excitation from air turbulence and control
forces.

Although Hansen et al. did not directly estimate an aerodynamic damping ratio, it is straightforward to compute the
contribution from aerodynamic damping, when other damping sources are known and subtracted from the estimated modal
damping. A method that evaluates the aerodynamic damping directly was recently proposed by Chen et al. (2017) and is a
wavelet-based linearization method. It also only requires long time series of the wind turbine under operating conditions.
Hansen et al. concludes that the SSI method provides better results than the exciter method to extract modal damping. Using
the SSI method showed a constant decrease for the modal damping for the first side-side tower mode for increasing wind
speed, while the modal damping for the first fore-aft tower mode is scattered about a mean of almost 13 % with a standard
deviation of 1.3 %. The SSI method was also used by Kramers et al. (2016) to obtain modal damping for fore-aft and side-
side modes of a 3.6 MW OWT under idling conditions. Damping ratios for these modes under idling conditions were only
around 3.0 %, since aerodynamic damping contributes only little to the overall damping. Damping ratios in the same range
were obtained from the measurement campaign performed at the Belwind offshore wind farm, which consists of 55
monopile-based 3.0 MW wind turbines. For turbines in parked conditions and subject to higher wind speeds (10 – 15 m/s),
damping ratios with a median of around 2.0 % and 3.0 % were found for the first fore-aft and first side-side mode,
respectively. The next three modes dominant for the response of the OWT (second fore-aft, second side-side with nacelle
component, and second fore-aft with nacelle component) had a significantly smaller damping ratio with a median smaller
than 2.0 % (Devriendt et al., 2014). Data from the same measurement campaign, but for wind turbines during power
production were published by Weijtjens et al. (2014). Damping ratios of the first fore-aft as well as the first side-side mode
showed a continuous increase when plotted against the wind speed. The median of the damping ratio for both modes starts at
a value of 2.0 % for the smallest wind speed (1.6 m/s) and goes up to almost 8.0 % and 3.0 % for the first fore-aft and side-
side mode, respectively. The measured data were later compared with results from a time-domain simulation (Shirzadeh et
al., 2014). Simulation and measurements did not match very well and the authors concluded that this is likely caused by not
accurately accounting for hydrodynamic forces.

A comparison between measurement and simulation data using the same analysis tool as Shirzadeh et al. (2014) for the time-
domain simulation was also conducted for an onshore wind turbine (Ozbek and Rixen, 2013). For this study, test campaigns
were performed for operational and parked conditions on a 2.5 MW wind turbine and the dynamic response of the structure



was monitored by using conventional strain gauges, photogrammetry, and laser interferometry. An operational modal analysis algorithm was used to obtain damping ratios for tower modes. The results for the first fore-aft and side-side mode were in the same range as the damping ratios obtained from simulations of the wind turbine in time-domain. The authors also compare their results with Hansen et al. (2006) and neither the trend nor the range matches for damping ratios of tower

modes.

To summarize, the methods to derive aerodynamic damping values are abundant as researchers working on this topic. Values derived by empirical methods and analytical formulations matches in most of the cases for wind speeds below its rated value, but show somewhat different behaviour for above rated wind speeds. Even for studies performed for wind turbines of similar size the damping values for higher wind speeds various widely. Although these differences were already shown in early

publications, none of the studies aimed to evaluate the accuracy of using a linear damper to represent aerodynamic damping. This paper does, therefore, not introduce a new method to derive an aerodynamic damping ratio, but presents an empirical study that determines the optimal damping coefficient for linear damping at tower top representing the dynamic interactions between rotor and support structure. This gives rise to the main question sought answered in this work, i.e. how accurate does a linear damper account for dynamic interactions between rotor and support structure? In order to investigate this,

different rotor load models and combinations of linear dampers are investigated and compared to the response of a support structure from dynamic analyses under combined aerodynamic and hydrodynamic loading. Results show that reduced rotor load models do not capture the dynamics of the structure accurately enough and that analytically derived formulas underestimate the damping ratio for aerodynamic damping above rated wind speed.

**2 Integrated and decoupled dynamic analysis**

The nomenclature on integrated, coupled, decoupled and other methods for load simulations and analysis of OWTs is not unified. In fact, the term "integrated analysis" has been used with widely varying meaning (c.f. Seidel et al., 2005; Kaufer et al., 2009). In this study, integrated analysis refers to an aero-hydro-servo-elastic load simulation of the entire OWT (support structure and rotor are modelled numerically) that is performed in a single time-domain simulation. The OWT is subject to combined wind and wave excitation and neither the model of the support structure nor the rotor are simplified or reduced

(Fig. 1).

It is not mandatory to perform the integrated analysis with one software package; indeed a combination of two simulation tools, an aerodynamic solver coupled with general structural analysis software, as described in Kaufer et al. (2009), can be considered as an integrated analysis under the definition given in this paper. In literature this analysis method is often called a coupled or fully-coupled analysis. Since OWTs are subject to nonlinear and time-history dependent effects coming from

wind and wave excitation, the integrated or coupled analysis is currently considered as the most accurate method for the dynamic analysis of OWTs. However, an integrated analysis is usually computationally demanding and requires detailed data of the support structure and turbine.




A computationally more efficient analysis method is the decoupled analysis. Here, the rotor in the numerical model is replaced by force and momentum time series, referred to as rotor loads, acting on tower top (Fig. 1). Rotor loads are typically precomputed or generated by means of simplified rotor load models, such as classical models based on thrust coefficients, spectral or stochastic models (Muskulus, 2015a). Precomputed rotor loads are usually obtained from an aero-

elastic simulation of a standalone turbine fixed at turbine bottom or from a wind turbine with non-moving, rigid support structure (Fig. 1). It is important to use a rigidly clamped rotor, since load time series from a simulation with moving support structure will lead to resonance problems (Muskulus, 2015b). Rotor load time series are then extracted from turbine bottom and applied at the top of the tower to perform a dynamic analysis of the support structure using standard finite-element or multi-body software. The rotor-nacelle-assembly in a decoupled analysis is usually simplified. Often an equivalent rotor-

nacelle-assembly is modelled as a lumped mass on top of the tower with a mass moment of inertia (Ong et al., 2017; van der Male and Lourens, 2015; Schløer et al., 2016). In this case, rotor loads must not include forces or moments from gravitational loads.

Rotor load time series used with decoupled models are often reduced to a point force and bending moment acting on tower top, in order to further simplify and accelerate the dynamic analysis of the support structure. The point force is applied in the

direction of the wind and perpendicular to the rotor (here x-axis) and represents the thrust force acting on the support structure. The bending moment, representing torque, is commonly applied around the axis in side-side direction of the turbine (i.e. parallel to the rotor plane and perpendicular to the direction of the wind; y-axis in Fig. 1). These loads are considered having the main impact on the dynamic response of the wind turbine and simplified rotor load models often do not provide a model for forces and moments acting in the remaining directions. A decoupled analysis that is not using force

and moment time series in all six degree of freedom (full rotor loads) is referred to as a reduced rotor load model in this paper.

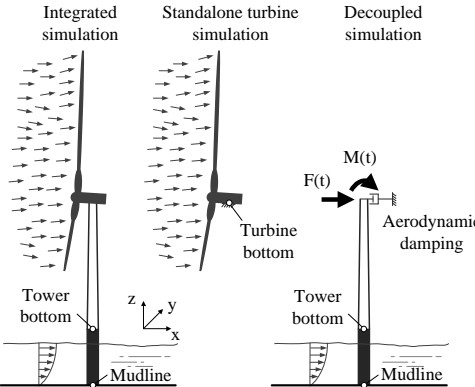

**Figure 1: Offshore wind turbine and simulation methods**



As already emphasized in the previous section, decoupling the rotor loads from the dynamic analysis of the support structure neglects important dynamic interactions between rotor and support structure. In an integrated simulation, the tower top motion reduces aerodynamic loads when it moves in the direction of the wind and increases the loads on the support structure vice versa. This motion mainly affects the thrust force. Using the assumption of an ideal rotor it is possible to

derive a simple one-dimensional model for estimating the thrust force, $F_T$, at tower top (Hansen, 2008)

$$F_T = \frac{1}{2} C_T \rho_a A_R U^2 \tag{1}$$

where $C_T$ is the thrust coefficient depending on the current state of the rotor, $\rho_a$ the air density, $A_R$ the rotor disc area, and U the wind speed. The latter can be divided in two components, when a turbulent wind field is considered

$$U = \overline{U} + u \tag{2}$$

where $\overline{U}$ is the mean and u the turbulent wind speed. The model can easily be extended to account for tower top motions in wind direction by subtracting the velocity of the tower top in wind direction, $\dot{x}$, from the wind speed, U, thereby calculating a

relative velocity for the rotor.

$$U_{rel} = \overline{U} + u - \dot{x} \tag{3}$$

Using the relative velocity and the model from Eq. 1, the thrust force can be separated into a static mean thrust force, $F_{T_{mean}}$, a dynamic turbulent thrust force, $F_{T_{turb}}$, and a dynamic damping force, $F_{damping}$.

$$F_T = \frac{1}{2} C_T \rho_a A_R \overline{U}^2 + \frac{1}{2} C_T \rho_a A_R (u^2 + 2\overline{U}u) + \frac{1}{2} C_T \rho_a A_R (\dot{x}^2 - 2u\dot{x} - 2\overline{U}\dot{x}) = F_{T_{mean}} + F_{T_{turb}} + F_{damping} \tag{4}$$

Rotor load models already account for the contribution of all three components, but precomputed rotor loads from a standalone turbine simulation only include the mean and the turbulent thrust force. The damping force has to be applied in

addition, when precomputed rotor loads are used in a decoupled analysis. Tower top velocities, $\dot{x}$, and turbulent wind speed, u, are small compared to the mean wind speed, $\overline{U}$. Hence, the first ($\dot{x}^2$) and second term ($2u\dot{x}$) of the damping force is neglected and the estimation of the damping force reduces to

$$F_{damping} = -C_T \rho_a A_R \overline{U}\dot{x} = -c_{AD}\dot{x} \tag{5}$$

where $c_{AD}$ describes the damping coefficient of a viscous damper representing the aerodynamic damping force. Assuming a thrust coefficient constant for distinct mean wind speeds, the coefficient, $c_{AD}$, is a function only of $\overline{U}$.

Aerodynamic damping is often expressed in terms of damping ratios. The damping ratio is defined as the ratio of the damping coefficient to the system's critical damping, $c_c$, and is calculated as follows





$$\zeta_{AD} = \frac{c_{AD}}{c_c} \tag{6}$$

with

$$c_c = 2m\omega \tag{7}$$

where m is the effective mass at tower top and $\omega$ the damped frequency.

## 3 Method

In this study the damping coefficients for the viscous damper applied on tower top are not calculated analytically, but
obtained by matching the response from a decoupled with an integrated simulation. According to guidance in current
standards (DNVGL-ST, 2016), the aerodynamic damping of the turbine will come out right performing an integrated
simulation for the dynamic analysis of the OWT. In order to enable a fair comparison the OWT in the integrated as well as
the decoupled simulation was subject to identical environmental conditions. This requires that the precomputed rotor loads
that were applied at tower top of the decoupled model, were generated using the same turbulent wind input file as for the
integrated simulation of the OWT. Furthermore, the wave spectrum and the random seed used for generating the wave
elevation time series for the decoupled model were identical to the one for the integrated model. Integrated as well as
decoupled simulations were performed for operational load cases based on DLC 1.2 (DNVGL-RP, 2016). The load
combinations that are part of DLC 1.2 are the main contribution to the fatigue lifetime of an OWT. A continuous 60-minute
period (excluding transients) is simulated per wind speed to ensure statistical reliability, as prescribed in standards (IEC,
2009; DNVGL-RP, 2016). Environmental data for wind and wave excitation are taken from the lumped scatter diagram for
the K13 shallow water side described in the UpWind design basis (Fischer et al., 2010).
The viscous damping coefficients are numerically optimized to match the desired output location, that is either the variance
of tower top displacements, the variance of forces and moments acting on tower bottom or an equivalent fatigue load (EFL)
from force and moment time series at tower bottom. Ideally, one would like to match the displacements at tower top for each
time step and each degree of freedom. Obtaining a perfect match for tower top displacements would be equal to a
substructuring technique thereby leading to an identical response in the support structure. However, decoupled models do not
always allow for matching tower top displacements due to non-zero shifts in mean or missing excitation frequencies that
significantly contribute to the deflection of the tower. Hence, the variance of tower top displacements is often used to
compare and evaluate a simplified analysis model (Schløer et al., 2016).
The tower bottom as an output location was chosen since it is an important location for fatigue checks in conceptual design
phases and parameter studies. Moreover, it often serves as an interface between support structure designer and wind turbine
supplier to exchange displacement or load time series during the design process.





An EFL provides a first rough estimate of the fatigue damage and basically describes a constant-amplitude load range that would cause an equivalent amount of damage as the original variable-amplitude load time series. It is calculated as

$$EFL = \left( \sum_{i=1}^{N} \frac{S_i^{\,m}}{N} \right)^{1/m} \tag{8}$$

where N is the number of load cycles applied, $S_i$ is the load range amplitude at cycle i, and m is a material parameter, which is chosen to match the properties of welded steel for the support structure. This simple description of fatigue damage
accumulation is based on Wöhler's equation (SN-curve), which assumes that each cycle of a constant load range amplitude causes a particular amount of damage, and that this damage is proportional to the load range amplitude raised to the power of m. Among the cycle counting techniques, the rainflow-counting algorithm (Amzallag et al., 1994) has been shown to match experimental results well and was used to determine the number of cycles, N, and the corresponding load amplitudes, S.

Plotting the ratio between the decoupled and integrated simulation over the viscous damping coefficient for variance or EFL
showed that the ratio, in case that only one single viscous damper in wind direction is applied, is a monotonically decreasing function of the damping coefficient. Hence, Brent's method (Press et al., 1994) was utilized for finding the damping coefficient for the ratio 1.0, that expresses a perfect match between integrated and decoupled analysis.

For decoupled models with more than one viscous damper at tower top, the objective of the optimization is defined as minimizing the residual of the ratio between the response from the decoupled and the integrated model. In case that the
optimization aims to match the variance of tower top displacements the function to minimize is given as

$$LMS_{TT} = \sqrt{ \sum_{i=1}^{6} \left( \frac{VAR_i^{DC}}{VAR_i^{IS}} - 1 \right)^2 } \tag{9}$$

where $VAR_i^{DC}$ and $VAR_i^{IS}$ is the variance of tower top displacements from the decoupled (DC) and integrated (IS) simulation in one out of the six degrees of freedom, i. The objective function describes thereby the least mean square (LMS) of the variance of tower top displacements. Six variables have to be optimized simultaneously and due to the non-linear optimization problem and the strong coupling between the damping coefficients a stochastic search method is used to
minimize the objective function. This does not guarantee that the global minimum is found, but it is sufficient for this study, since this paper does not aim to provide a methodology to obtain aerodynamic damping coefficients but aims to evaluate the accuracy and limitations of using decoupled simulations and linear dampers to represent the interaction between support structure and wind turbine rather than determining generally valid damping coefficients.

A genetic algorithm similar to the implementation in Schafhirt et al. (2014) is utilized for the optimization. The biological
terminology used to briefly describe the main features of the algorithms follows the description in Holland (1975), where also more details about optimization with genetic algorithms can be found. In this study the damping coefficients were




binary-coded and put together into a string with a length of 76 bits, representing a so-called chromosome. A population size of 15 individuals was chosen and considered sufficient, since the probability that every point in the search space is reachable from the initial population only by crossover already exceeds 99.5% (Reeves and Rowe, 2003). The initial population was randomly generated and loss of diversity was chosen as the stop criteria. A linear fitness scaling function was used and a

mutation rate of 0.05 was chosen for the mutation process. This algorithm is computationally demanding since around hundred generations have to be simulated until the algorithm converges and the optimization usually has to be performed several times with different initial conditions in order to increase the likelihood that a global optimum has been found (Arora, 2012). However, a genetic algorithm is straightforward in implementation and simple to adjust for the desired optimization problem.

**4 The offshore wind turbine model**

Results presented in this paper are based on simulations with the generic OWT used within Phase I of the OC3 project (Jonkman et al., 2010). The OWT consists of the NREL 5 MW reference turbine (Jonkman et al., 2009) and a monopile support structure located in a water depth of 20 m. The 30 m monopile has a constant diameter of 6 m and a constant thickness of 0.06 m, while the tower mounted on top of the monopile has a linearly tapered diameter and thickness with a top

diameter of 3.87 m and a top thickness of 0.019 m. The OWT has a hub height of 88.15 m. The turbine has a cut-in wind speed of 3 m/s and a cut-out wind speed of 25 m/s. Load simulations were therefore performed for 11 different wind speeds ranging from 4 m/s to 24 m/s with a step size of 2 m/s. The rated wind speed for this turbine is 11.4 m/s. Figure 1 illustrates the model of the OWT.

The model is implemented in the flexible multibody simulation tool Fedem Windpower (Version R7.2.1, Fedem Technology

AS) in order to perform integrated dynamic analyses in time-domain. Two different implementations are used for this study. The first one, referred to as the integrated model, performs the dynamic analysis of the OWT under combined aerodynamic and hydrodynamic loading. The second implementation is the so-called decoupled wind turbine model, since the simulation of the rotor loads are decoupled from the dynamic analysis of the support structure. The same numerical model is used for the second implementation, but aero-elastic simulations of the wind turbine are switched of. Instead aerodynamic loads are

applied with force and momentum time series acting on tower top. These time series were precomputed from an aero-elastic simulation of a standalone turbine model clamped on turbine bottom (see Fig. 1). These rotor loads do not include gravitational forces, since the numerical model of the OWT uses the detailed representation of the rotor-nacelle-assembly (identical to the integrated model). This also ensures that differences to the integrated analysis will not be caused by a different representation of the rotor-nacelle-assembly. The dynamic interactions between rotor and support structure, mainly

the aerodynamic damping, is represented in the decoupled model by a discrete dashpot at tower top. This study investigates four different types of decoupled models: A decoupled model with thrust force applied on tower top and one single viscous damper in wind direction (DC I), with thrust and torque applied on tower top and one single viscous damper in wind



direction (DC II), with full rotor loads and one single viscous damper in wind direction (DC III), and with full rotor loads and viscous dampers in all degrees of freedom at tower top (DC IV).

**5 Decoupled models with reduced rotor loads**

The simplest dynamic analysis of an OWT using decoupled models is the implementation of the support structure in finite-
element or multi-body software and the application of thrust time series as defined in section 2 on tower top (Abhinav and Saha, 2015; van der Male and Lourens, 2015). This simulation model is called the decoupled model DC I in this study. The aerodynamic damping for this type of decoupled model is represented by a viscous damper acting in the direction of the wind and perpendicular to the rotor (here x-axis). The viscous damping coefficient is numerically optimized to match the integrated simulation regarding (1) the variance of tower top displacements in x-direction, (2) the variance of the overturning
moment around the y-axis on tower bottom, and (3) the equivalent fatigue load of the overturning moment around the y-axis on tower bottom using Brent's method as described in section 3.

The coefficients found by the algorithm for the three above mentioned response locations were used to calculate a modal damping ratio for the first fore-aft mode, which is plotted for DC I against the wind speed in Fig. 2. The figure shows that the empirically derived damping ratios increase almost continuously for increasing wind speeds. Exceptions can be observed
for wind speeds around the rated wind speed of the turbine.

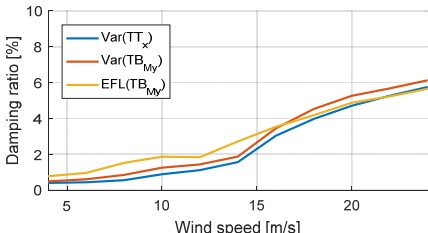

**Figure 2: Damping ratios optimized for different output for DC I. The three different curves show the damping ratio for matching the variance of the tower top displacement in x-direction (Var(TT$_x$)), the variance of the tower bottom bending moment around the y-axis (Var(TB$_{My}$)), and the EFL of the tower bottom bending moment around the y-axis (EFL(TB$_{My}$)).**

It is also interesting to compare the response on other output locations and directions for these empirically derived damping coefficients. Figure 3 and 4 show the ratio between the response from the decoupled and the response from the integrated simulation, for displacements at tower top, the variance of forces and moments at tower bottom and the equivalent fatigue loads from forces and moments at tower bottom for DC I and two different wind speeds (below rated wind speed (8 m/s) and above rated wind speed (20 m/s)).



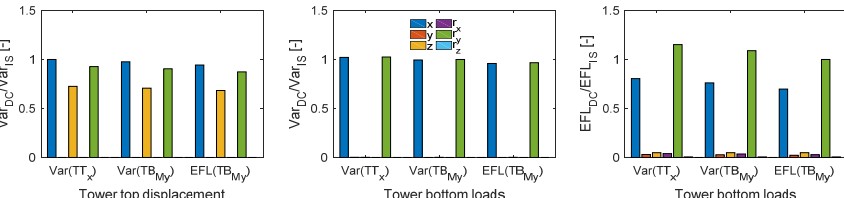

**Figure 3: Response for tower top displacements and tower bottom loads for DC I and a wind speed of 8 m/s. Damping coefficients matching the variance of the tower top displacement in x-direction (Var(TT$_x$)), the variance of the tower bottom bending moment around the y-axis (Var(TB$_{My}$)), and the EFL of the tower bottom bending moment around the y-axis (EFL(TB$_{My}$)) are chosen for**
**the decoupled analysis. Each bar stands for a displacement or force/moment in one out of the six degrees of freedom.**

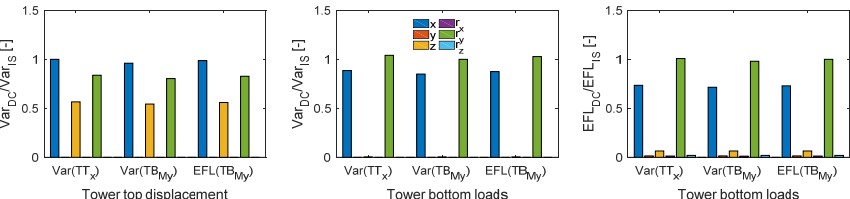

**Figure 4: Response for tower top displacements and tower bottom loads for DC I and a wind speed of 20 m/s.**

It can be seen that the variance for the translational tower top displacement in y-direction and the rotational displacements in x- as well as z-direction are highly underrepresented. The height of the bars for these output directions is within the line
thickness of the plot. The same counts for the variance of forces in y- and z-direction as well as moments around the x- and z-axes at tower bottom. Only the equivalent fatigue loads for these output locations are slightly visible.

Moreover, the figures show that the response from a decoupled simulation does not exceed the response from an integrated simulation. This is an expected result, since for DC I only a thrust force is applied. This also shows that for this type of decoupled simulation only a single viscous damper in wind direction is necessary. Adding more dampers would only
decrease the response further.

This leads to the decoupled model DC II that is often used in studies and in the conceptual design phase. For this model a bending moment around the y-axis is applied on tower top in addition to the already applied thrust force (Ong et al., 2017; Schløer et al., 2016). The bending moment is supposed to represent the torque as explained in section 2. Again, the damping coefficient for the single viscous damper in wind direction was optimized to match the variance at tower top and tower
bottom and the equivalent fatigue load at tower bottom, respectively. The calculated damping ratios for the first fore-aft mode are plotted against the wind speed in Fig. 5.





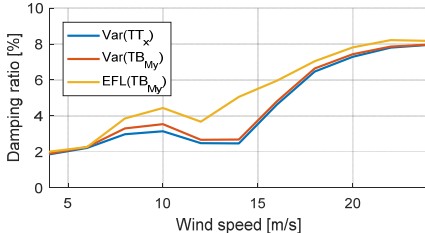

**Figure 5: Damping ratios optimized for different output for DC II.**

An increase of the damping ratio for higher wind speeds can be observed and is similar to results obtained for DC I, but results for wind speeds around rated wind speed differ somewhat. The damping ratios are generally higher compared to DC

5   I, which is due to the additional moment acting on tower top.

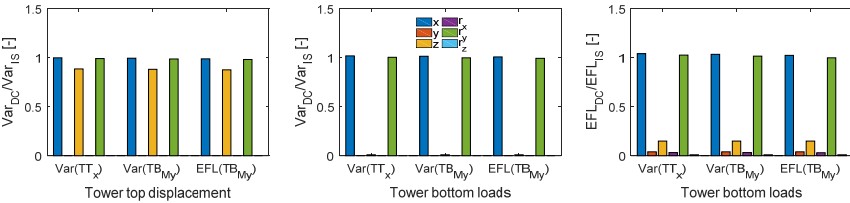

**Figure 6: Response for tower top displacements and tower bottom loads for wind speed of 8m/s and DC II.**

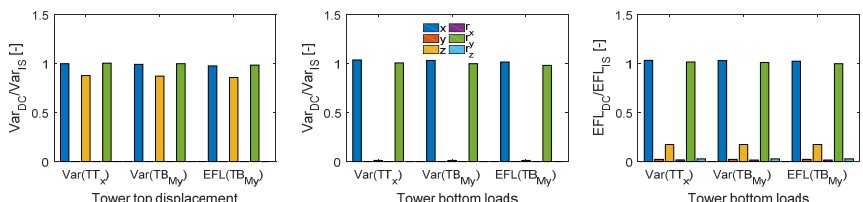

**Figure 7: Response for tower top displacements and tower bottom loads for wind speed of 20m/s and DC II.**





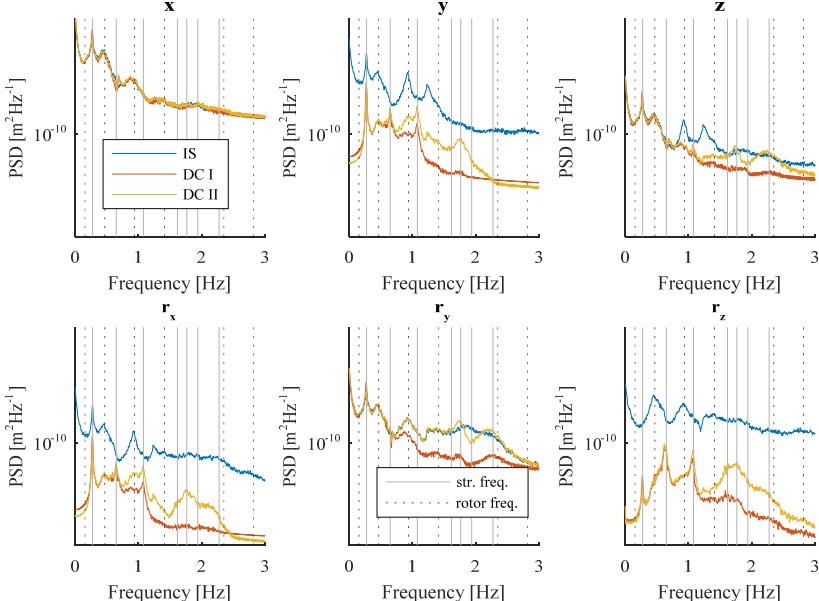

**Figure 8: PSD for tower top displacements of DC I and DC II with a wind speed of 8 m/s. Continuous vertical lines show structural eigenfrequencies, while vertical dotted lines (··) show rotational frequencies of the rotor.**



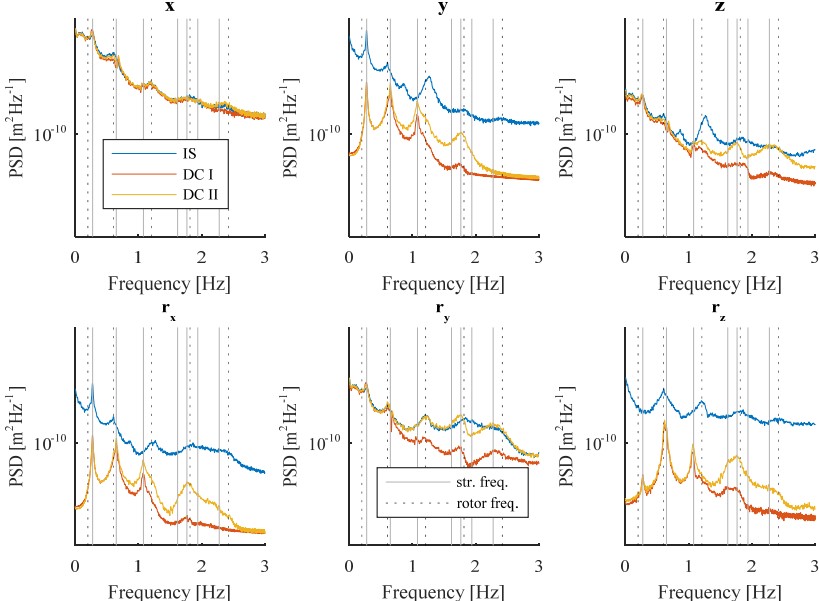

**Figure 9: PSD for tower top displacements of DC I and DC II with a wind speed of 20 m/s.**

Figure 6 and 7 evaluates the response on tower top and tower bottom for the numerically derived damping coefficients for DC II for a wind speed of 8 m/s and 20 m/s. It can be seen that for all three optimization objectives (variance at tower top,

variance at tower bottom, and equivalent fatigue load at tower bottom), the displacements and forces in x-direction and the displacements and moments around the y-axis matches the integrated simulation better than for the decoupled model DC I. In contrast to that, the response at other output locations is again highly underrepresented.

An explanation can be found when looking at the power spectral density (PSD) for tower top displacements. Figure 8 and 9 display spectra for all six degrees of freedom at tower top for DC I and DC II with a damping coefficient numerically

optimized for the variance of tower top displacements in x-direction and for a wind speed of 8 m/s and 20 m/s, respectively. The spectra for optimized variance and equivalent fatigue load at tower bottom show a similar behaviour and spectra are, therefore, only plotted for the optimization of variance at tower top in the further course of this paper. While the spectrum for displacements in x-direction and the rotational displacements around the y-axis for DC II fairly match, spectra for the four remaining directions show big differences. The spectra show that the energy content for the decoupled models is

somewhat smaller than the one from the integrated simulation. The largest energy content is still in x-direction, but it seems that rotor loads in other directions cannot be neglected if the response of the decoupled model shall better match the integrated simulation. The figures also show that peaks between decoupled and integrated model typically occur at different





frequencies. For example peaks in DC I and DC II are clearly visible in all spectra at a frequency of 1.07 Hz (one of the structural eigenfrequencies), but do not occur in spectra of the integrated model. On the contrary, the integrated model shows peaks at rotational frequencies coming from the rotor (e.g. at the 6P frequency of 1.21 Hz for a wind speed of 20 m/s). This observation applies likewise for DC I and DC II and for several structural and rotational eigenfrequencies. In order to include

the rotational frequencies, decoupled models with rotor loads acting in all directions at tower top are considered in this study. These models are referred to as decoupled models with full rotor loads.

## 6 Decoupled models with full rotor loads

The implementation of the support structure in finite-element or multi-body software is the same as the previously introduced decoupled models with reduced rotor loads. The single difference is that all six force and moment time series

extracted from the standalone turbine simulation are applied on tower top. For the first decoupled model with full rotor loads a single viscous damper in wind direction is added on tower top (DC III), representing the aerodynamic interaction between turbine and support structure. A single damper in wind direction is chosen, since aerodynamic damping mainly damps the fore-aft motion of the wind turbine. As for DC I and DC II, Brent's method was utilized to optimize the damping coefficient with the objective to match the variance for tower top displacements in x-direction and the variance and equivalent fatigue

load for the overturning moment ($M_y$) at tower bottom. Results are show in Fig. 10.

**Figure 10: Damping ratio over wind speed for DC III and DC IV.**

The curve closely matches the one for the decoupled model with reduced rotor loads (DC II). However, calculating the ratios for displacements, forces and moments at tower top and tower bottom shows a different behaviour now (Fig. 11 and 12).





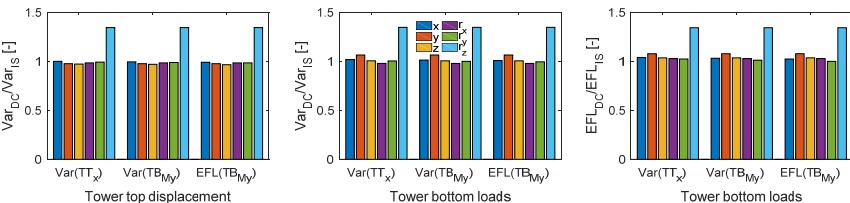

**Figure 11: Response for tower top displacements and tower bottom loads for wind speed of 8m/s and DC III.**

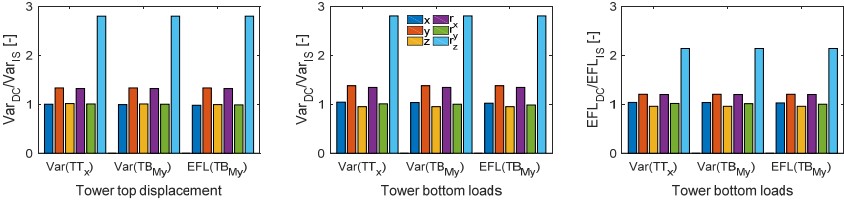

**Figure 12: Response for tower top displacements and tower bottom loads for wind speed of 20m/s and DC III.**

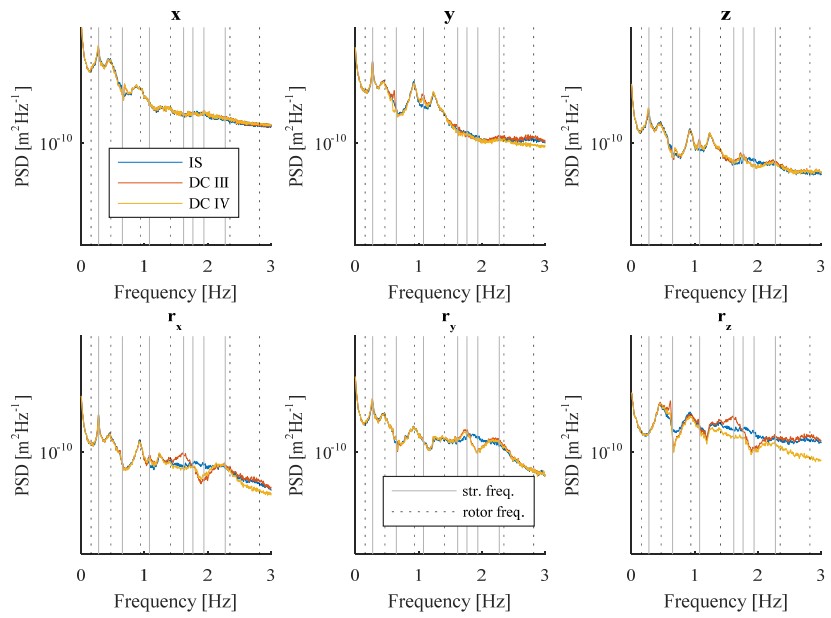

**Figure 13: PSD for tower top displacements of DC III and DC IV with a wind speed of 8 m/s.**



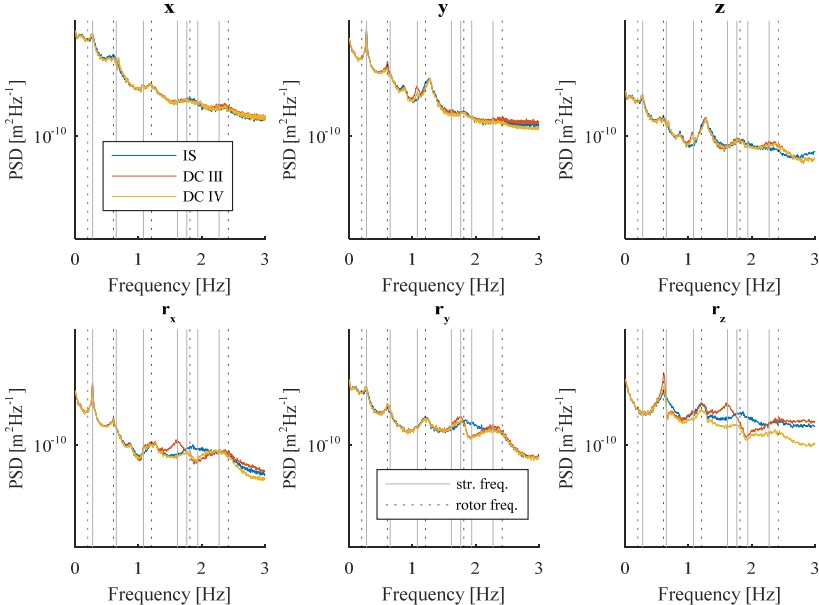

**Figure 14: PSD for tower top displacements of DC III and DC IV with a wind speed of 20 m/s.**

The results of the decoupled model (DC III) are not conservative anymore. In fact, the ratio for displacements for bending moments around the y-axis are up to four times higher for a wind speed of 12 m/s (not shown) and still almost three times

higher for a wind speed of 20 m/s.

The PSD for tower top displacements in rotational z-direction ($r_z$) in Fig. 14 explains this difference. The peak at the rotational 3P frequency (0.6 Hz) is clearly larger for DC III compared to the integrated model. Structural eigenfrequencies at 1.07 Hz and 1.61 Hz for translational displacements in z-direction and rotational displacements in x direction, respectively, show larger differences to the integrated model likewise. Hence, additional damping is required in order to match the results

of the integrated simulation. It seems that especially the torsion (rotational movement around the vertical axis) of the OWT has to be damped.

The fourth decoupled simulation that is analysed (DC IV) uses the precomputed rotor loads in all degrees of freedom on tower top (as DC III) and has in addition six viscous dampers in all translational and rotational directions at tower top. The aim was to adjust the damping coefficients for all six viscous dampers in such a way that the decoupled simulation model

DC IV matches the variance at tower top in all six degrees of freedom of the integrated simulation. For this purpose the sum of the squared residuals (Eq. 9) is minimized using a genetic algorithm as described in section 3.




The optimization was performed for wind speeds of 8 m/s, 12 m/s, 16 m/s, and 20 m/s. Results in Fig. 15 and 16 show the ratio between the decoupled simulation using DC IV and the integrated simulation for variance of tower top displacements and the variance as well as the EFL for forces and moments at tower bottom in all six degrees of freedom for two wind speeds. The first plot (variance at tower top) shows that the genetic algorithm found a fair match for the displacements at

tower top. However, also the variance and EFL at tower bottom differ only around 5% with the exception of the moment around the vertical axis where the decoupled simulation underestimates the EFL by around 7.5%.

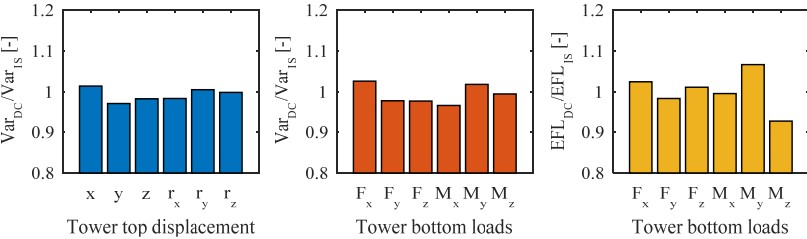

**Figure 15: Response of tower top displacements and tower bottom loads for wind speed of 8m/s and DC IV.**

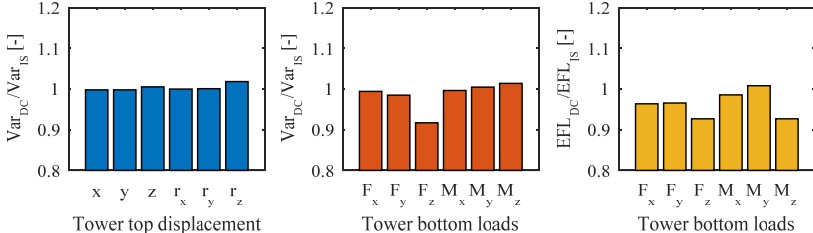

**Figure 16: Response of tower top displacements and tower bottom loads for wind speed of 20m/s and DC IV.**

The PSD in Fig. 13 and 14 confirm this observation. Spectra for DC IV show a better match with the integrated model compared to the previous decoupled models. However, differences in energy content for different frequencies still occur.

Finally, the damping coefficients for the linear damper in wind direction that were obtained from the optimization with the genetic algorithm for the four wind speeds under consideration, are used to calculate a damping ratio for the first fore-aft

mode. The damping ratios are plotted together with the damping ratios over the wind speeds for DC III in Fig. 10. It can be seen that the damping ratios are slightly smaller due to more damper acting at tower top, but the damping ratios nevertheless increase further above rated wind speed.

The last results presented in this paper are the power spectral density for forces in x- and y-direction and the moment around the y-axis at tower bottom (Fig. 17 and 18). These output locations are select for a comparison among decoupled and

integrated models, since they significantly contribute to the overall response at tower bottom (see Tab. 1 for a comparison of mean and standard deviation of forces and moments in all six degrees of freedom for the integrated analysis and wind speeds of 8 m/s and 20 m/s).




**Table 1: Mean and standard deviation (STD) of forces and moments at tower bottom from integrated analysis.**

| Wind speed | Quantity | $F_x$ | $F_y$ | $F_z$ | $M_x$ | $M_y$ | $M_z$ |
|---|---|---|---|---|---|---|---|
| 8 m/s | Mean | 386.53 kN | -5.61 kN | -5778.42 kN | 2.59 MNm | 30.84 MNm | 0.01 MNm |
| | STD | 75.72 kN | 7.14 kN | 15.24 kN | 0.67 MNm | 5.97 MNm | 0.92 MNm |
| 20 m/s | Mean | 318.33 kN | -27.31 kN | -5768.11 kN | 6.64 MNm | 27.08 MNm | 0.24 MNm |
| | STD | 81.68 kN | 32.94 kN | 29.00 kN | 2.52 MNm | 5.86 MNm | 1.75 MNm |

The spectra in Fig. 17 and 18 confirm the observations made before. Decoupled models with reduced rotor loads (DC I and DC II) exhibit a lack of of rotational frequencies and show their peaks at mainly structural eigenfrequencies. In addition, their energy content is generally smaller due to fewer loads acting on tower top. The response from decoupled models with full rotor loads match the integrated simulation better with DC IV showing the best agreement with respect to the integrated model (e.g. at structural eigenfrequencies of 1.07 Hz and 1.61 Hz – visible for the force in y-direction).

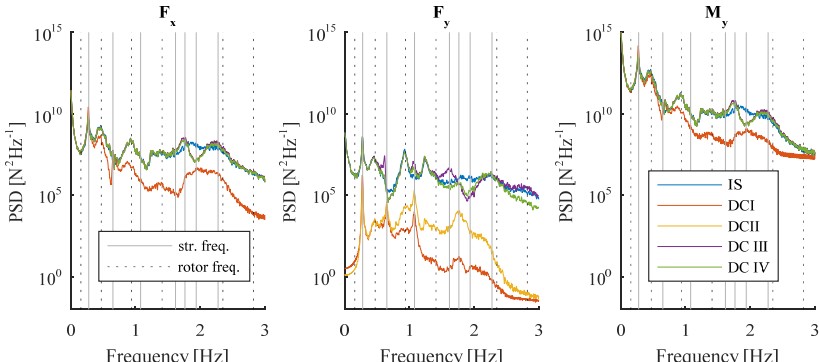

**Figure 17: PSD for forces and moment at tower bottom for wind speed of 8 m/s**

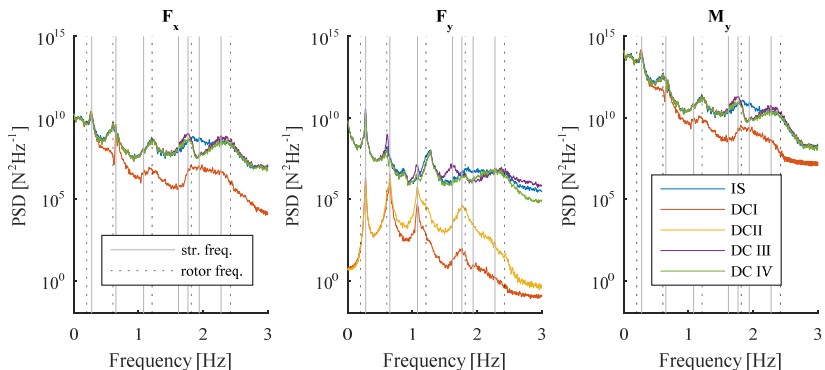

**Figure 18: PSD for forces and moment at tower bottom for wind speed of 20 m/s.**



## 7 Discussion

Results presented in the previous section show that it is possible to match the response of an integrated simulation at distinct output locations and particular directions with decoupled models. This is even possible with the simplest model (DC I), where only a thrust force is applied at tower top and a single linear damper in wind direction is chosen to represent the

dynamic interactions between rotor and support structure. However, the response at other output locations or directions can strongly deviate from an integrated model. This was the case for both decoupled models, where only reduced rotor loads were applied (DC I and DC II). The reason is clear when looking at the PSD for these models. The response for these models is dominated by structural eigenfrequencies. These frequencies occur in the response of integrated models as well, but they are, apart from the first structural eigenfrequency, not as dominant as for the decoupled models with only thrust and torque

applied. It is also visible in the plots of the PSD that the first structural eigenfrequency is less dominant for higher wind speeds (e.g. 20 m/s). This might explain why a single viscous damper applied in wind direction provides fair results for wind speeds below rated. For these wind speeds, the fore-aft motion is governing the response of the OWT.

The response of the integrated simulation for all wind speeds includes rotational frequencies of the rotor that partially have a larger impact on the response than the structural eigenfrequencies. This was visible through all of the PSDs shown in this

paper. Hence, it is reasonable to apply rotor loads in all degrees of freedom at tower top in order to better match the response of an integrated simulation. The decoupled model with full rotor loads and one single viscous damper in wind direction (DC III) showed generally a better match than DC I and DC II when comparing their PSD with an integrated simulation. However, for a few output locations, the response became non-conservative and sometimes up to four times higher compared to the integrated analysis. Examination of the PSD showed that higher energy content mainly at structural

eigenfrequencies is the reason for it. In order to compensate for this, translational and rotational dampers in all six degrees of freedom have to be applied at tower top, as it has been done for DC IV. This enables to match the most dominant frequencies, but higher frequencies still show deviations to the integrated simulation. Matching the higher frequencies will not be possible with a single linear damper.

The application of translational and rotational dampers in all six degrees of freedom together with full rotor loads leads to a

fair match for output locations considered in this study with differences at the most of 7.5 %. This might be an answer for the time being to the question raised in the introduction, how accurate a linear damper can represent the dynamic interaction between rotor and support structure. However, the method utilized in order to obtain these results is not generally applicable to all support structures. Lattice support structures, such as jackets, are more complex and the responses at tubular joints, which are locations prone to fatigue, are not only driven by the fore-aft motion of the turbine (Popko et al., 2013). Generally

it has to be kept in mind that the optimization of damping coefficients presented in this paper is only performed for one single monopile based OWT. Furthermore, only one set of load cases and only uni-directional wind and waves are considered. Results might differ for variations in environmental conditions or changes in the support structure design.





## 8 Conclusion

Wind turbine simulations with four different decoupled models using precomputed rotor loads were performed for this study. The decoupled models differ in number of rotor load time series applied on tower top and the number of viscous dampers representing the dynamic interaction between support structure and rotor. Results of tower top displacements and forces and

moments at tower bottom were compared with an integrated simulation which is considered as providing accurate results. The damping coefficients for the viscous dampers were numerically optimized to match variance or EFL at tower top and tower bottom. Three conclusions regarding the use of decoupled simulations for the dynamic analysis of support structures can be drawn:

**The damping ratio for a viscous damper in fore-aft direction further increases for wind speeds above its rated value.**

Analytically derived formulas to obtain the damping ratio for the fore-aft movement of a wind turbine show that the damping ratio stays constant or even slightly decreases for wind speeds above rated. However, the results presented in this paper show that for the OWT considered, the damping ratio constantly increases with higher wind speeds. This conclusion can already be found in Salzmann and van der Tempel (2005). It suggests that formulas presented to calculate aerodynamic damping for wind turbines do not adequately cover the entire range of dynamic interactions between rotor and support structure. It is

thereby clear that using an engineering approach of 5% of critical damping for all wind speeds (van der Tempel, 2000) or formulas that predict such an uniform damping ratio above rated wind speed (Salzmann and van der Tempel, 2005) underestimate the damping caused by the rotor and should not be used when the simulation of the wind turbine aims to provide results as accurate as a decoupled simulation allows.

**Depending on the desired output, a decoupled model using precomputed rotor loads from a fixed wind turbine**

**simulation and a single damper in wind direction might be sufficient.** In fact, it is possible to perfectly match an integrated simulation for distinct output locations (e.g. overturning moment at tower bottom or variance of tower top displacements in fore-aft direction) with a decoupled simulation and only one single damper in fore-aft direction, but results at other output locations are highly underestimated when using reduced rotor loads (DC I and DC II) or did not show a clear trend when full rotor loads were applied at tower top (DC III). The choice to use a decoupled model for the dynamic analysis

might be reasonable when indeed only output at a single location is required, such as the overturning moment at tower bottom or mudline, that can be a design criterion during a conceptual design phase or for a parameter study. Especially for a monopile based OWT with larger diameter where the response due to wave excitation is governing and the side-side motion is little, compared to the fore-aft motion of the turbine, a decoupled model might provide sufficiently accurate results. This will be different for jacket based OWT, where wave excitation is little, due to the lattice structure with small diameter tubes,

and the importance of more than the overturning moment (e.g. also torsion) that will influence the axial force and out-of-plane and in-plane bending moment at jacket joints.

**The decoupled model with full rotor loads and six dampers in translational as well as rotational displacements at tower top provided the best match for an integrated simulation.** A fair fit for damping coefficients was found for the four





different wind speeds that were considered in this study (below rated as well as above rated wind speed) and results in terms of variance at tower top and variance and EFL at tower bottom differed at most by 7.5%. This is a significant improvement compared to the results of other decoupled models, where the ratio between decoupled and integrated response were sometimes around 4.0 or close to zero. However, the method to obtain the damping coefficients for all six viscous dampers

used in this paper is computationally demanding and impractical for industrial application. Hence, a more efficient method or formula to obtain optimal damping coefficients for viscous dampers is desirable. Moreover, since it is clear that the dynamic interaction between rotor and support structure cannot be perfectly represented by simple linear damping, a more sophisticated damping model that also damps higher frequencies is required.

### Acknowledgments

Support by the Norwegian Research Centre for Offshore Wind technology (NOWITECH FME, Research Council of Norway, contract no. 193823) is gratefully acknowledged.

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
