# Peer review of "Decoupled simulations of offshore wind turbines with reduced rotor loads and aerodynamic damping"

_Wind Energy Science, 2017_

## Referee Comment (RC1) · T. A. Nygaard (Referee) · 17 Sep 2017

comments1_nygaard_wes2017_29.pdf contains the general and specific comments

wes-2017-29_tan1.pdf contains some further (and overlapping) specific comments, and the technical corrections

Please also note the supplement to this comment: https://www.wind-energ-sci-discuss.net/wes-2017-29/wes-2017-29-RC1-supplement.zip

---

## Referee Comment (RC2) · Anonymous Referee #2 · 17 Oct 2017

In this paper, decoupled simulation methods with reduced and full rotor loads are compared to an integrated simulation. The paper does not introduce a new method to derive an aerodynamic damping ratio, but presents an empirical study that determines the optimal damping coefficient for linear damping at the tower top.

For offshore wind turbines on monopile support structures, the soil-structure interaction can significantly affect the performance of the system. Please clarify if the soil-structure interaction is considered in this study.

The choice of two wind speeds (i.e. 8m/s and 20m/s) in the case studies should be justified.

It would be appropriate to add a case study to validate the calculated damping ratio.

---

## Author Comment (AC1) · 14 Nov 2017

**Response to Referee #1 (Tor Anders Nygaard)**

**Discussion Paper wes-2017-29**

**Decoupled simulations of offshore wind turbines with reduced rotor loads and aerodynamic damping**

*Sebastian Schafhirt and Michael Muskulus*

Dear Tor Anders Nygaard,

We thank you for your careful work and interest in our manuscript. It is nice to see that you address topics that we have thought about or examined during the preparation of the manuscript as well as new aspects/points of view. Please find the response to your comments in blue below.

With best regards,

Also on behalf of my co-author

Sebastian Schafhirt

(1)    The article compares computational effort for several models, but I suspect there are different levels of parallelization at play. If this is the case, please mention that more specifically. I would like to see apples-to apples comparison of computational speed for the FEDEM model with the decoupled rotor model and pre-computed time-series of rotor forces, relative to the complete integrated FEDEM model. The effort for pre-computation of forces and optimization of the damping coefficients do not need to be included here. Does the relative difference change if this is run on a single processor?

The reviewer's assumption is correct. Simulations in Fedem Windpower were run on a workstation that allows for parallel computing. The workstation contains 16 CPUs and usually 15 simulations were run in parallel. This set-up does not slow down the speed for a single time-domain simulation. A single simulation on a single processor requires the exact same time. The simulation time itself depends on the hardware used in order to run the simulation. The integrated simulations with a simulated time of 4000 seconds performed for the study presented in the paper required a CPU time of 57 minutes on average. This matches the elapsed time (wall clock time), since results were not written to a hard disc and only saved in RAM. This very efficient simulation set-up is described in detail in Zwick's doctoral thesis from 2015 (Daniel Zwick, Simulation and Optimization in Offshore Wind Turbine Structural Analysis, NTNU, 2015). The thesis also provides a script to handle the exchange of result files among RAM and HDD.

The decoupled simulations were performed with exactly the same set-up. A decoupled analysis of 4000 second took around 13 minutes.

The authors thought about include this comparison prior the submission of the manuscript, but decided not do to so due to (1) the strong dependence of the time needed to perform a simulation on the actual hardware used, (2) the fact that a general FEM software might perform a decoupled simulation even faster than Fedem, and (3) the opportunity to use faster simulation methods (e.g. Impulse Based Substructuring in combination with parallel computing on Graphics Processing Units).However, we are happy to include a small paragraph about the computational performance when requested.

(2)    New sequence for the paper: From complex to simple models.

The structure of the manuscript as it is submitted to the journal is in fact not the initial structure of the paper. We changed the sequence to start with simple models and end with a more complex model in order to explain our motivation to look at more complex models. There would have been no point to investigate more complex models in case that the simple models (DCI or DCII) showed already accurate results. Hence, we started with the simplest and most commonly used decoupled models in academic studies and preliminary design phases and showed the differences to an integrated/fully coupled simulation. The subsequent models are used to investigate how accurate a decoupled model with linear damper can match the integrated analysis. The most complex model presented in the study is in addition certainly not the most complex decoupled model. Models with quadratic damping or a damping force that depends on a function can possibly come even closer to an integrated simulation.

The authors agree that changing the sequence of the paper is an option, but would suggest slightly emphasizing the motivation instead to justify the current sequence of the model presented.

(3)   The optimization problem is here defined to adjust the six damping coefficients to match the fully integrated model by minimizing eq. 9. In eq. 9, the variance residuals of the 6 tower top deflections have equal weight, although they have different influence on the von Mises stress on the different parts of the structure. As mentioned earlier in the article, this is OK if a perfect match is found for all DOFs, but this is not always possible. Also, two time series of deflections may be very different, but have the same variance. Please comment if other optimization criteria have been tested as well, such as fatigue life due to all loads, in in selected hot spots, or important stress components at selected hot spots.

That is a very good point and we looked indeed at a wide range of optimization criteria before performing this study. Potential optimization criteria, apart from von Mises stress and the already mentioned criteria in the manuscript (equivalent fatigue lifetime and variance) are an overall fatigue lifetime of the support structure, matching the cycle distribution after rainflow-counting, and stress calculation for particular hot spots. These optimization criteria caused additional challenges and we decided to use rather simple criteria. Our decision was motivated by (1) the fact that more detailed postprocessing might not allow to explore the reason for differences (EFLs per force/moment are more accurate than an total EFL) and (2) since we only used unidirectional wind and waves the response is mainly dominated by loads in wind direction. We intended to have a look at the simplest models and optimization criteria. As mentioned before, if even the simplest comparison does not show good agreement, a future investigate will not make sense. Furthermore, it is current industry practice to use for example an overturning moment at seabed as a design criterion in preliminary design phases (Seidel et al., State-of-the-art design processes for offshore wind turbine support structures, Stahlbau, Volume 85, 2016). We agree with the reviewer that it is important to consider wind and wave misalignment and keeping track of effective stress, but we consider this as material for a further study or an extension of the one presented here.

The similarities of response time series is ensured by having visual checks of the spectra – not only the area, but also the shape of the spectra should be considered – and comparing the cycle distribution after performing the counting algorithm. Both has been done during the study and spectra are show in the paper itself.

(4)   The comparisons of PSD plots of forces in the article is a very powerful tool for explaining the differences in the results, although the logarithmic scale mask some of the differences. The way the study is set up, also gives the opportunity to look at comparison examples in the time domain, since a perfect decoupled model and detailed rotor model in theory should give the same time history of loads. Please show examples of comparison. I would like to see 100s of each of the six load components at the tower root comparing 1) the integrated simulation, 2) the decoupled model with full rotor loads and six optimized dampers, and if possible, 3) the decoupled model with full rotor loads and no dampers, at 8 m/s, rated wind speed and 20 m/s.

We agree with the reviewer and we will provide the requested plot in the revised version of the manuscript.

(5)   In the EFL comparisons of figures 15 and 16, please include a comparison of "total" EFL, based on von Mises stress, on the upwind side of the tower at the tower root and just below the nacelle.

The comparison of a total EFL for the requested locations will be included in the revised version of the manuscript.

(6)    In the conclusions, it is stated that "the method to obtain the damping coefficients for all six viscous dampers used in this paper is computationally demanding and impractical for industrial applications. Hence, a more efficient method or formula to obtain optimal damping coefficients for viscous dampers is desirable". I think one reason the optimization algorithm is demanding, is that we are trying to optimize one "convoluted" objective (eq. 9) by varying six coefficients simultaneously. Maybe the starting point for the optimization could be improved by tuning one and one or two and two coefficients by looking at outputs more directly linked to those coefficients. For example, select the damping coefficient for x motion, Cx to match variance of My at the tower bottom, Cy to match the variance of Mx at the tower bottom, set Cz equal to Cy (should be the same, except for the influence of shear and rotor tilt), and Crz to match the variance of torsion shear stress at the tower top. If approaches like this are already used, or tested and discarded, please comment on this.

The approach described by the reviewer has in fact be used before. In the course of the study several decoupled models were investigated (see also point 11). Some of the models included two dampers (e.g. one in fore-aft and a second one in side-side direction). For these models the successive tuning of damping coefficients is very efficient. However, it became more complicated with three and more dampers on tower top due to the strong coupling among the degrees of freedom. Damper in a certain direction does not only affect the damping in its own direction. We also calculated sensitivities and tried to optimize the damping coefficients. None of the approaches led to the results obtained by the genetic algorithm. We, therefore, decided to employ the algorithm for the model with six dampers on tower top.

We propose to include a short sentence in the relevant chapter to mention that a successive optimization approach was tested and can be used if only two dampers are added.

(7)    How different are Cx found in DC 1 through DC 4?

A plot showing damping ratios for all four models over the wind speed will be included in the paper.

(8)    On page 15, line 8, the full and decoupled models are the same as above, except that only one damper is applied in the x direction at the tower top for the decoupled model. This coefficient is tuned to match 1) tower top variance of displacement, 2) the variance of the overturning moment, and 3) the equivalent fatigue load for the overturning moment (My) at tower bottom. In figure 10, the damping coefficient optimized for matching EFL is very different from the damping coefficient optimized for matching tower bottom bending moment variance. Please examine why the results differ so much, because this would help us understand more of the challenges using simplified criteria such as tower top deflections.

Figure 10 shows the damping ratios for the decoupled model (DC III) with full rotor loads applied and one damper in wind-direction on tower top. The damping coefficient is optimized for three different objective functions. The damping ratios obtained to match variances (tower top and tower bottom) show smaller damping ratios for the wind speeds 12 m/s 14m/s and 16m/s. A similar trend can be observed for DC II (Figure 5), while the damping ratios obtained for DC I (Figure 2) show a more or less similar behavior for all three objective functions. The differences between the decoupled models (DC I – DC III) are the loads applied on tower top. It seems that adding an overturning moment at tower top to an already applied thrust force affects variance and EFL at tower bottom differently. There is not an obvious reason that explains these differences. However, it has to be taken into account that

controller activities significantly influence the response for the above mentioned wind speeds and that EFLs are non-linear. In fact only a few cycles with lower amplitude can significantly decrease the EFL. Comparing the cycle-counting between a model optimized to match variance at tower bottom and a model optimized to match EFL at tower bottom shows a higher damaged for the variance-optimized model. The higher damaged is caused by only a few cycles with a slightly smaller amplitude (mainly in the lower third of the maximum cycle range). We agree with the reviewer that it is of importance to understand how simplified criteria can be used in the load analysis. We guess that it will become a broader topic and that it might be too much to investigate it in this study. We propose to add a paragraph in the discussion part – since it is an important observation – and leaf it for future work.

(9)   One point is that even if two models have the same variance of tower bottom bending moments, the EFL may be different due to the strong nonlinearity from loads to EFL (eq. 8). Then it is surprising that we seem to get identical results in the figures 11 and 12, for the three optimization methods, since the decoupled rotor model has been run with very different damping. Please double-check that the figures 11 and 12 are correct and comment on this. In the beginning of the article it is stated that the aerodynamic damping is very important for fatigue, but many of the comparisons show very small differences for significant differences of damping levels.

Very good observation. We checked the figure mentioned by the reviewer again and they are plotted correctly. Differences between the three optimized models exist, but these are small for the two wind speeds shown and are hardly to distinguish considering the scale of the vertical axis. We propose to change the scale to make the small differences more visible.

(10)  On page 17, line 3, it is stated that "The results of the decoupled model (DC 3) are not conservative anymore. In fact, the ratio for displacements for bending moments around the y-axis are up to four times higher for a wind speed of 12 m/s (not shown) and still almost three times higher for a wind speed of 20 m/s." I think it should be bending moments around the z-axis. [Correct] Please clarify the use of "conservative" or simply state whether the loads for model DC3 are higher or lower than the integrated model. When I hear that a model is "conservative", I assume it has higher loads than the reference.

Many thanks for the comment. We will correct the typo and rephrase the paragraph in the revised version of the manuscript. It will clearly be stated and we will get rid of the term "conservative".

(11)  On the same page, it is stated that "It seems that especially the torsion (rotational movement around the vertical axis) of the OWT has to be damped." I think this is a very important comment. In my opinion, the next logical step is a model DC3b with dampers in tower top x translation, and tower top z rotation. It is even possible that the optimization could be carried out first as in DC3 (tuning only coefficient Cx), and then, keeping Cx constant, tune Crz. Even if the two coefficients after being initialized separately have to be fine-tuned simultaneously, should this give a significant speedup compared to model DC4. Would model DC3b then perform almost as good as model DC4? What would be the increase in computational cost over model DC3 (due to optimizing two damping coefficients vs. one)? How sensitive is the life-time fatigue at the tower root and tower top due to the torsion shear stress? Please do this, or comment if this has been tested and discarded already.

We appreciate the reviewer's thoughts. As mentioned above (point 6) not all of the decoupled models and optimization criteria investigated in this study are presented in the

paper. We reduced the number of models to keep the manuscript interesting for the reader without reducing the main conclusion of the study. Our motivation is to assess the accuracy of decoupled models used in studies and preliminary design phases, show that aerodynamic damping is non-linear and that it increases with higher wind speeds. The proposed model DC 3b has not been used in studies before and will not perform as well as DC IV. The latter one can be said since the genetic algorithm would find a solution with damping coefficients set to zero that is performing better than the current solution for DC IV.

It is correct that finding the damping coefficients for Cx and Crz is significant faster than optimizing damping coefficients in six degree of freedom using a genetic algorithm. The results are comparable to the model DC III with the exception that the displacement and moments around the vertical axis matches the results from an integrated analysis. We intentionally did not include the model in the current manuscript, but can provide results or add it as an additional decoupled model in Chapter 6.

(12)     Chapter 5 – Decoupled models with reduced rotor loads: I think this section should examine or give more details of the cited articles on the importance of the different force and moment components on the life time fatigue at the hot spots in the tower root and top. This can be achieved in several ways, e.g.:

1. Re-run the integrated model through the fatigue load cases, and remove one force/moment component from the rotor at the time (except rotor thrust), or

2. Remove one stress component at the time at selected hot spots at the tower root and tower top.

3. Supplement the figures 3, 4, 6, 7, 11, 12 , 15 and 16 with "total" EFL based on all stress components, at the upwind side of the tower (since all load cases here have wind and waves aligned).

The reviewer brings up an interesting topic that is not been covered in the manuscript yet. Performing the proposed steps will give a better picture on which rotor loads have to be applied to obtain accurate results with decoupled models. The authors examined the rotor loads at the beginning of the study. Mean and variance of rotor loads (forces and moments at turbine bottom from a standalone rotor simulation) and the tower top displacements of an integrated simulation were plotted against the wind speeds (see figure below). Even if there is not a direct dependence between variance and EFLs, it can be concluded that with higher variance, EFL will increase as well. On the other hand, increasing only the mean of a rotor load will lead to the same EFL. Looking at the plots will thereby giving a first brief idea which rotor loads matter for the load analysis. Force or moment time series with little variance, even if their mean is increasing for higher wind speeds, will only marginal impact the EFL.

As mentioned earlier in this rebuttal, our motivation is to present a simple investigation first and continue with more complex models and evaluation methods, if it is of interest. The reviewer suggests a quite comprehensive study and we assume that it rather content for further studies/work.

[Figure]

(13)    Model DC 1 applies only rotor thrust to the tower top. This model is attractive for users without access to a detailed rotor model, since overall CT curves are often available, and the time series can be pre-computed from eq. 1. Model DC2, including tower top overturning moment, requires a detailed rotor model. This step up in complexity relative to model DC1 is maybe a bit understated. Once we have the six force/moment components available from a detailed rotor model, the extra effort over model DC 2 is to optimize the additional damping coefficients, for each wind speed bin.

Very good point. We appreciate the comment and will emphasize it in the revised version of the manuscript.

(14)    Comment Page 8: heavy sentence!

So, the objective function is matching tower top variance of deflection, or EFL, and a one-dimensional search was used to find the damping coefficient.

But the way you have set up this optimization, a perfect match would give identical time series of loads, right? On the other hand, two very different time series can have the same variance. Are there other objective functions that could explore the consistency of this optimization setup even better?

Please mention if other objective functions have been tested, and why this one was chosen.

The perfect match as it is used in this study refers to matching the variance (or EFL) of displacements, forces or moments of an integrated analysis exactly. It does not mean that the time series itself will match exactly. This is generally not possible as long as only a thrust force or overturning moment is applied at tower top. We agree that time series can be very different even if the variances matches. However, examining the cycle-counting and the PSD ensure that the response time series are similar. The revised manuscript will also include a plot of the response in time-domain, as suggested by the reviewer. The authors tested and used other objective functions, as mentioned in point 6. We tried to match the cycle-distribution of the rainflow-counting algorithm by using the principles of distance optimization. Furthermore, we aimed to optimize the RMS of the displacement time series, which was not successful due to a non-zero shift in the mean of the response time series and small differences in frequency and phase. Since these approaches did not work out, we limited our objective function to an easy expression: the variance and EFL calculation for distinct output locations. We used in total ten output locations and presented three of them (1) variance of tower top displacements, (2) variance of tower bottom loads, and (3) EFL of tower bottom loads.

(15)    Remaining comments/sticky notes in the actual script:

We read the comments and will address them accordingly in the revised manuscript. The meaning of the highlighted sentence in the abstract is not totally clear to the authors. We assume that it is mistakenly highlighted.

---

## Author Comment (AC2) · 14 Nov 2017

**Response to Referee #2 (anonymous)**

**Discussion Paper wes-2017-29**

**Decoupled simulations of offshore wind turbines with reduced rotor loads and aerodynamic damping**

*Sebastian Schafhirt and Michael Muskulus*

Dear reviewer,

We thank you for your careful work and dedicated effort. Your comments are much appreciated. We will revise our manuscript based on your comments. Details on how we are going to address them can be found below.

With best regards,

Also on behalf of my co-author

Sebastian Schafhirt

(1) For offshore wind turbines on monopile support structures, the soil-structure interaction can significantly affect the performance of the system. Please clarify if the soil-structure interaction is considered in this study.

The reviewer's statement is correct. The model as it is defined in Phase I of the OC3 project is used. This model does not include a soil model. We will add this piece of information in beginning of Chapter 4. However, adding soil-structure interaction will not change the outcome of this study.

(2) The choice of two wind speeds (i.e. 8m/s and 20m/s) in the case studies should be justified.

In total 11 wind speeds between cut-in and cut-out wind speed are investigated. The rated wind speed for the turbine that is subject to this thesis is at 11.4 m/s. We decided to present results for one wind speed below rated wind speed and for one wind speed above rated wind speed. This decision is made due to differences in controller activities below and above rated wind speed. Aerodynamics differ in the different regimes. This is stated in the manuscript (page 10, last paragraph). We propose to rephrase the paragraph in order to clearly justify our choice.

(3) It would be appropriate to add a case study to validate the calculated damping ratio.

A validation is certainly a good point, but a little bit out of the scope for this work. The wind turbine subject to this study is a generic turbine and measurement data are not available. The study as presented in this manuscript can only conclude that the model is not linear. A valid question would be to evaluate how linear the aerodynamic damping is in reality. We can add a corresponding comment in the conclusion if requested.

---

## Author Response (AR1)

**Response to Referee #1**
**(Tor Anders Nygaard)**

**Discussion Paper wes-2017-29**

**Decoupled simulations of offshore wind turbines with reduced rotor loads and aerodynamic damping**

*Sebastian Schafhirt and Michael Muskulus*

Dear Tor Anders Nygaard,

We thank you once more for your careful work and interest in our manuscript. We revised the manuscript based on your comments. Please find our point-by-point reply below.

With best regards,

Also on behalf of my co-author

Sebastian Schafhirt

(1)    The article compares computational effort for several models, but I suspect there are different levels of parallelization at play. If this is the case, please mention that more specifically. I would like to see apples-to apples comparison of computational speed for the FEDEM model with the decoupled rotor model and pre-computed time-series of rotor forces, relative to the complete integrated FEDEM model. The effort for pre-computation of forces and optimization of the damping coefficients do not need to be included here. Does the relative difference change if this is run on a single processor?

The introduction (Chapter 1) includes an additional paragraph in order to address the topic of reduction in computational costs and our reasons to not compare it in the study (please see page 4, line 17-24).

(2)    New sequence for the paper: From complex to simple models.

We agree that changing the sequence of the paper is an option, but decided to stick with our choice. However, we adjusted the introduction to justify the current sequence of the model presented (cf. page 4, line 17-24).

(3)    The optimization problem is here defined to adjust the six damping coefficients to match the fully integrated model by minimizing eq. 9. In eq. 9, the variance residuals of the 6 tower top deflections have equal weight, although they have different influence on the von Mises stress on the different parts of the structure. As mentioned earlier in the article, this is OK if a perfect match is found for all DOFs, but this is not always possible. Also, two time series of deflections may be very different, but have the same variance. Please comment if other optimization criteria have been tested as well, such as fatigue life due to all loads, in in selected hot spots, or important stress components at selected hot spots.

That is a very good point and we looked indeed at different optimization criteria. We included a new paragraph in Chapter 3 (cf. page 8, line 15-20).

(4)    The comparisons of PSD plots of forces in the article is a very powerful tool for explaining the differences in the results, although the logarithmic scale mask some of the differences. The way the study is set up, also gives the opportunity to look at comparison examples in the time domain, since a perfect decoupled model and detailed rotor model in theory should give the same time history of loads. Please show examples of comparison. I would like to see 100s of each of the six load components at the tower root comparing 1) the integrated simulation, 2) the decoupled model with full rotor loads and six optimized dampers, and if possible, 3) the decoupled model with full rotor loads and no dampers, at 8 m/s, rated wind speed and 20 m/s.

We agree with the reviewer. The requested plot is included in the revised version of the manuscript.

(5)    In the EFL comparisons of figures 15 and 16, please include a comparison of "total" EFL, based on von Mises stress, on the upwind side of the tower at the tower root and just below the nacelle.

Only variance and EFLs are compared within the study, while other results might be possible to compare as well (e.g. rainflow-counting and von Mises stress). We decided to keep it consistent and do not introduce a new value to compare towards the end of the manuscript. Therefore, a total EFL based on von Mises stress is not included in Figure 15 and 16.

(6)    In the conclusions, it is stated that "the method to obtain the damping coefficients for all six viscous dampers used in this paper is computationally demanding and impractical for industrial applications. Hence, a more efficient method or formula to obtain optimal damping

coefficients for viscous dampers is desirable". I think one reason the optimization algorithm is demanding, is that we are trying to optimize one "convoluted" objective (eq. 9) by varying six coefficients simultaneously. Maybe the starting point for the optimization could be improved by tuning one and one or two and two coefficients by looking at outputs more directly linked to those coefficients. For example, select the damping coefficient for x motion, Cx to match variance of My at the tower bottom, Cy to match the variance of Mx at the tower bottom, set Cz equal to Cy (should be the same, except for the influence of shear and rotor tilt), and Crz to match the variance of torsion shear stress at the tower top. If approaches like this are already used, or tested and discarded, please comment on this.

The approach described by the reviewer has in fact be used in the course of this study. We added a paragraph in Chapter 3 (cf. page 9, line 4-7)

(7)     How different are Cx found in DC 1 through DC 4?

A plot showing damping ratios for all four models over the wind speed is included in the revised manuscript (cf. Figure 19).

(8)     On page 15, line 8, the full and decoupled models are the same as above, except that only one damper is applied in the x direction at the tower top for the decoupled model. This coefficient is tuned to match 1) tower top variance of displacement, 2) the variance of the overturning moment, and 3) the equivalent fatigue load for the overturning moment (My) at tower bottom. In figure 10, the damping coefficient optimized for matching EFL is very different from the damping coefficient optimized for matching tower bottom bending moment variance. Please examine why the results differ so much, because this would help us understand more of the challenges using simplified criteria such as tower top deflections.

Figure 10 shows the damping ratios for the decoupled model (DC III) with full rotor loads applied and one damper in wind-direction on tower top. The damping coefficient is optimized for three different objective functions. The damping ratios obtained to match variances (tower top and tower bottom) show smaller damping ratios for the wind speeds 12 m/s 14m/s and 16m/s. A similar trend can be observed for DC II (Figure 5), while the damping ratios obtained for DC I (Figure 2) show a more or less similar behavior for all three objective functions. The differences between the decoupled models (DC I – DC III) are the loads applied on tower top. It seems that adding an overturning moment at tower top to an already applied thrust force affects variance and EFL at tower bottom differently. There is not an obvious reason that explains these differences. However, it has to be taken into account that controller activities significantly influence the response for the above mentioned wind speeds and that EFLs are non-linear. In fact only a few cycles with lower amplitude can significantly decrease the EFL. Comparing the cycle-counting between a model optimized to match variance at tower bottom and a model optimized to match EFL at tower bottom shows a higher damaged for the variance-optimized model. The higher damaged is caused by only a few cycles with a slightly smaller amplitude (mainly in the lower third of the maximum cycle range). We agree with the reviewer that it is of importance to understand how simplified criteria can be used in the load analysis. We guess that it will become a broader topic and that it might be too much to investigate it in this study.

(9)     One point is that even if two models have the same variance of tower bottom bending moments, the EFL may be different due to the strong nonlinearity from loads to EFL (eq. 8). Then it is surprising that we seem to get identical results in the figures 11 and 12, for the three optimization methods, since the decoupled rotor model has been run with very

different damping. Please double-check that the figures 11 and 12 are correct and comment on this. In the beginning of the article it is stated that the aerodynamic damping is very important for fatigue, but many of the comparisons show very small differences for significant differences of damping levels.

Very good observation. We checked the figure mentioned by the reviewer again and they are plotted correctly. Differences between the three optimized models exist, but these are small for the two wind speeds shown and are hardly to distinguish considering the scale of the vertical axis. Hence, we changed the scale of Figure 11 and 12.

(10)   On page 17, line 3, it is stated that "The results of the decoupled model (DC 3) are not conservative anymore. In fact, the ratio for displacements for bending moments around the y-axis are up to four times higher for a wind speed of 12 m/s (not shown) and still almost three times higher for a wind speed of 20 m/s." I think it should be bending moments around the z-axis. [Correct] Please clarify the use of "conservative" or simply state whether the loads for model DC3 are higher or lower than the integrated model. When I hear that a model is "conservative", I assume it has higher loads than the reference.

Many thanks for the comment. We corrected the typo and rephrased the paragraph (cf. page 1, line 16-18).

(11)   On the same page, it is stated that "It seems that especially the torsion (rotational movement around the vertical axis) of the OWT has to be damped." I think this is a very important comment. In my opinion, the next logical step is a model DC3b with dampers in tower top x translation, and tower top z rotation. It is even possible that the optimization could be carried out first as in DC3 (tuning only coefficient Cx), and then, keeping Cx constant, tune Crz. Even if the two coefficients after being initialized separately have to be fine-tuned simultaneously, should this give a significant speedup compared to model DC4. Would model DC3b then perform almost as good as model DC4? What would be the increase in computational cost over model DC3 (due to optimizing two damping coefficients vs. one)? How sensitive is the life-time fatigue at the tower root and tower top due to the torsion shear stress? Please do this, or comment if this has been tested and discarded already.

We appreciate the reviewer's thoughts. As mentioned above (point 6) not all of the decoupled models and optimization criteria investigated in this study are presented in the paper. We reduced the number of models to keep the manuscript interesting for the reader without reducing the main conclusion of the study. Our motivation is to assess the accuracy of decoupled models used in studies and preliminary design phases, show that aerodynamic damping is non-linear and that it increases with higher wind speeds. The proposed model DC 3b has not been used in studies before and will not perform as well as DC IV. The latter one can be said since the genetic algorithm would find a solution with damping coefficients set to zero that is performing better than the current solution for DC IV.

It is correct that finding the damping coefficients for Cx and Crz is significant faster than optimizing damping coefficients in six degree of freedom using a genetic algorithm. The results are comparable to the model DC III with the exception that the displacement and moments around the vertical axis matches the results from an integrated analysis.

(12)   Chapter 5 – Decoupled models with reduced rotor loads: I think this section should examine or give more details of the cited articles on the importance of the different force and moment components on the life time fatigue at the hot spots in the tower root and top. This can be achieved in several ways, e.g.:

1. Re-run the integrated model through the fatigue load cases, and remove one force/moment component from the rotor at the time (except rotor thrust), or

2. Remove one stress component at the time at selected hot spots at the tower root and tower top.

3. Supplement the figures 3, 4, 6, 7, 11, 12 , 15 and 16 with "total" EFL based on all stress components, at the upwind side of the tower (since all load cases here have wind and waves aligned).

The reviewer brings up an interesting topic that is not been covered in the manuscript yet. Performing the proposed steps will give a better picture on which rotor loads have to be applied to obtain accurate results with decoupled models. The authors examined the rotor loads at the beginning of the study. Mean and variance of rotor loads (forces and moments at turbine bottom from a standalone rotor simulation) and the tower top displacements of an integrated simulation were plotted against the wind speeds (see figure below). Even if there is not a direct dependence between variance and EFLs, it can be concluded that with higher variance, EFL will increase as well. On the other hand, increasing only the mean of a rotor load will lead to the same EFL. Looking at the plots will thereby giving a first brief idea which rotor loads matter for the load analysis. Force or moment time series with little variance, even if their mean is increasing for higher wind speeds, will only marginal impact the EFL.

As mentioned earlier in this rebuttal, our motivation is to present a simple investigation first and continue with more complex models and evaluation methods, if it is of interest. The reviewer suggests a quite comprehensive study and we assume that it rather content for further studies/work.

[Figure]

(13)     Model DC 1 applies only rotor thrust to the tower top. This model is attractive for users without access to a detailed rotor model, since overall CT curves are often available, and the time series can be pre-computed from eq. 1. Model DC2, including tower top overturning moment, requires a detailed rotor model. This step up in complexity relative to model DC1 is maybe a bit understated. Once we have the six force/moment components available from a detailed rotor model, the extra effort over model DC 2 is to optimize the additional damping coefficients, for each wind speed bin.

Very good point. We appreciate the comment and emphasized it in the revised version of the manuscript (cf. page 23, line 26-28).

(14)     Comment Page 8: heavy sentence!

So, the objective function is matching tower top variance of deflection, or EFL, and a one-dimensional search was used to find the damping coefficient.

But the way you have set up this optimization, a perfect match would give identical time series of loads, right? On the other hand, two very different time series can have the same variance. Are there other objective functions that could explore the consistency of this optimization setup even better?

Please mention if other objective functions have been tested, and why this one was chosen.

The perfect match as it is used in this study refers to matching the variance (or EFL) of displacements, forces or moments of an integrated analysis exactly. It does not mean that the time series itself will match exactly. This is generally not possible as long as only a thrust force or overturning moment is applied at tower top. We agree that time series can be very different even if the variances matches. However, examining the cycle-counting and the PSD ensure that the response time series are similar. The revised manuscript will also include a plot of the response in time-domain, as suggested by the reviewer. The authors tested and used other objective functions, as mentioned in point 6. We tried to match the cycle-distribution of the rainflow-counting algorithm by using the principles of distance optimization. Furthermore, we aimed to optimize the RMS of the displacement time series, which was not successful due to a non-zero shift in the mean of the response time series and small differences in frequency and phase. Since these approaches did not work out, we limited our objective function to an easy expression: the variance and EFL calculation for distinct output locations. We used in total ten output locations and presented three of them (1) variance of tower top displacements, (2) variance of tower bottom loads, and (3) EFL of tower bottom loads.

The corresponding paragraph has been revised accordingly (cf. Chapter 3).

(15)     Remaining comments/sticky notes in the actual script:

We read the comments and addressed them accordingly in the revised manuscript. We did not move the definition of critical damping in the introduction chapter, but pointed out that the damping ratios mentioned are provided in terms of the system's critical damping. The formula to calculate critical damping follows in Chapter 2.

The meaning of the highlighted sentence in the abstract is not clear to the authors. We assume that it is mistakenly highlighted.

**Response to Referee #2 (anonymous)**

**Discussion Paper wes-2017-29**

**Decoupled simulations of offshore wind turbines with reduced rotor loads and aerodynamic damping**

*Sebastian Schafhirt and Michael Muskulus*

Dear reviewer,

We thank you once again for your careful work and dedicated effort. Your comments are much appreciated. We revised our manuscript based on your comments. Details on how we addressed your comments can be found below.

With best regards,

Also on behalf of my co-author

Sebastian Schafhirt

(1) For offshore wind turbines on monopile support structures, the soil-structure interaction can significantly affect the performance of the system. Please clarify if the soil-structure interaction is considered in this study.

Correct. It is clarified now. (see 1st paragraph Chapter 4)

(2) The choice of two wind speeds (i.e. 8m/s and 20m/s) in the case studies should be justified.

We agree and added a brief explanation in the corresponding paragraph (page 11 line 10-12)

(3) It would be appropriate to add a case study to validate the calculated damping ratio.

A validation is certainly a good point, but a little bit out of the scope for this work. We included a comment in the discussion chapter (please see page 22 line 17 to 19)

**List of relevant changes**

**Discussion Paper wes-2017-29**

**Decoupled simulations of offshore wind turbines with reduced rotor loads and aerodynamic damping**

*Sebastian Schafhirt and Michael Muskulus*

All relevant changes made to the manuscript are highlighted. Additional figures are added (17, 18, and 19).

[revised manuscript text omitted]